# Stream of Search (SoS): Learning to Search in Language

**Kanishk Gandhi**[*]
Stanford University

**Denise Lee**
Stanford University

**Gabriel Grand**
MIT

**Muxin Liu**
Harvey Mudd

**Winson Cheng**
Stanford University

**Archit Sharma**
Stanford University

**Noah D. Goodman**
Stanford University

## Abstract

Language models are rarely shown fruitful mistakes while training. They then struggle to look beyond the next token, suffering from a snowballing of errors and struggling to predict the consequence of their actions several steps ahead. In this paper, we show how language models can be taught to search by representing the process of search in language, as a flattened string — a *stream of search (SoS)*. We propose a unified language for search that captures an array of different symbolic search strategies. We demonstrate our approach using the simple yet difficult game of Countdown, where the goal is to combine input numbers with arithmetic operations to reach a target number. We pretrain a transformer-based language model from scratch on a dataset of streams of search generated by heuristic solvers. We find that SoS pretraining increases search accuracy by 25% over models trained to predict only the optimal search trajectory. We further finetune this model with two policy improvement methods: Advantage-Induced Policy Alignment (APA) and Self-Taught Reasoner (STaR). The finetuned SoS models solve 36% of previously unsolved problems, including problems that cannot be solved by any of the heuristic solvers. Our results indicate that language models can learn to solve problems via search, self-improve to flexibly use different search strategies, and potentially discover new ones. [1]

> "To err is human, to backtrack divine"
> —Apocryphal

## 1 Introduction

Imagine, only ever seeing the right solutions to problems, never a mistake or recovery from it. You might learn that problems must be solved in one clean pass, rather than through exploration and error. Most data used to train language models (LMs) only reflects the outcome of a decision making process, not the process itself. LMs never learn to make mistakes. They never learn to search, plan or backtrack. Complex decision-making and reasoning requires search. In this paper we explore the impact of training a LM on the search process, including mistakes, and then allowing them to self-improve.

Transformer-based auto-regressive models have been shown to struggle with planning (Valmeekam et al., 2024; Pallagani et al., 2023; Momennejad et al., 2024). Recent work has highlighted this weakness in autoregressive models by identifying two main issues (LeCun, 2023; Bachmann & Nagarajan, 2024): 1) the snowballing of errors, where a single mistake can compound and lead to increasingly poor performance in subsequent steps (Ross et al., 2011; Arora et al., 2022), and 2) a difficulty in 'lookahead tasks', where the model must predict the consequences of its actions several steps ahead (credit assignment, Cf. Sutton & Barto, 2018). Both of these issues can be attributed to limited ability to search and backtrack. While recent efforts have combined language models with symbolic search algorithms (Ahn et al., 2022;

---

[*]Corresponding author: kanishk.gandhi@stanford.edu
[1]Code Available Here: https://github.com/kanishkg/stream-of-search

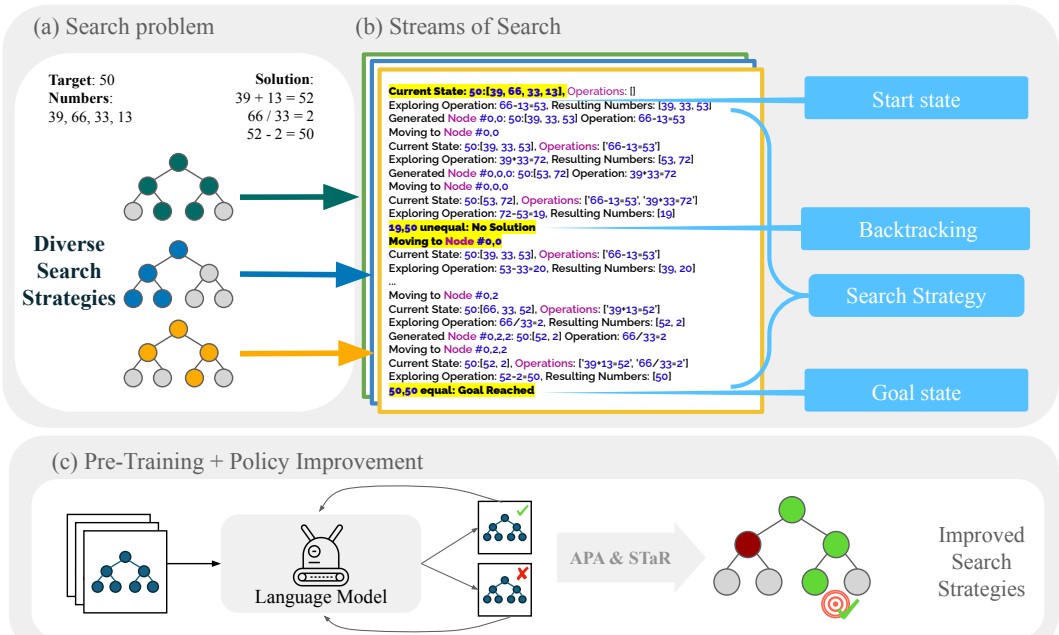

Figure 1: Overview of the Stream of Search (SoS) framework. (a) A search problem in Countdown is instantiated with input numbers and a target number. The input numbers need to be combined with simple arithmetic operations to get to the target. (b) The Stream of Search dataset contains search trajectories generated by diverse search strategies, including exploration and backtracking. (c) The language model is first trained on the SoS dataset and then iteratively improved using policy improvement techniques such as APA and STaR.

Yao et al., 2024) to mitigate some of these problems, they are limited — only supplementing language models during inference —they leave open the question of whether language models could effectively carry out search themselves. The most consequential outcome from learning to search could be during the *training* itself (Silver et al., 2018). If language models can learn to search during training, they may be able to discover more flexible search strategies through self-improvement. This could lead to models that are better equipped to handle the challenges posed by error compounding and lookahead tasks.

In this paper, we demonstrate that language models can be taught to search and backtrack in language, representing the process as a serialized string, a Stream of Search (SoS). We systematize the different components of search, such as exploration, backtracking, and pruning in a unified language. We instantiate this unified language in the context of a search problem inspired by the game of Countdown, a generalized version of the game of 24 (Yao et al., 2024; Countdown, 2024). A problem consists of input numbers and a target number. The goal is to combine the input numbers with arithmetic operations to reach the target (see Fig. 1a). Countdown presents a challenging search problem due to its high branching factor and the need to efficiently navigate the combinatorial search space towards the target number. We generate an initial training dataset of search trajectories or streams of search using different symbolic planners and simple heuristic functions, then train a language model on this diverse dataset. This can be contrasted with training on the optimal path once a solution is found, without the process of search and backtracking. We find that the Stream of Search LM substantially outperforms models trained to predict the optimal steps. Moreover, we observe improvements in the search and planning abilities of the Stream of Search model when finetuned to optimize for correctness using Advantage-Induced Policy Alignment (APA, Zhu et al., 2023) and expert iteration with STaR (Zelikman et al., 2022; Gulcehre et al., 2023).

Our results indicate that transformer-based language models, when shown how to recover from mistakes and search through different options, can learn to solve problems by searching. More importantly, our results indicate that these models can self-improve to autonomously

use different search strategies, solving previously unsolved problems. Finally, we see some evidence that they discover new search strategies when trained to optimize for accuracy.

## 2 Related Works

**Language models as components of a search system.** A line of recent work integrates language models as part of larger search and planning systems (Yao et al., 2024; Ahn et al., 2022). In these methods, LMs typically play two roles: (1) to generate candidate actions or successor states in the reasoning process, and (2) to evaluate proposed actions or states, by determining validity and/or assigning a heuristic value. A symbolic search algorithm such as BFS or DFS dictates the strategy for exploration, and how the steps or evaluators are called (Yao et al., 2024; Besta et al., 2023). While these methods have been shown to improve search accuracy on certain problems, the LM components are typically used only for inference, so their reasoning ability is not improved. In contrast, our work focuses on training LMs that are capable of exploration, backtracking, and other critical components of reasoning. Relative to these "extrinsic" methods, which use fixed search strategies, our method learns an "intrinsic" policy that allows the LM to autonomously search the solution space. In doing so, we avoid the high inference costs (Sel et al., 2023) required by tree-of-thoughts style approaches.

**In-context demonstrations of search.** In contrast to using separate symbolic search algorithms to guide search, in-context demonstrations can also be used to demonstrate search procedures in language (Gandhi et al., 2023; Sel et al., 2023). These methods allow the language model to perform tree search based on the demonstrated search procedures. While improving efficiency in search, these methods are restricted by the choices of the search procedure demonstrated in the in-context examples, such as the demonstrated exploration strategy or the search heuristic.

**Process supervision.** Process supervision of LMs is another method where an external verifier model is trained to learn the rewards of each intermediate reasoning step and the LM is then trained on this detailed feedback. Process supervision was shown to outperform outcome supervision in mathematical reasoning tasks (Lightman et al., 2023). However, the training of the verifier requires a very large labelled dataset with human-generated annotations for each intermediate step, which may not scale well for other, more complex domains. Comparatively, our method directly improves the model's planning and search abilities and removes the need to train a verifier reward model.

**Learning from search trajectories.** Similar to our work, Yang et al. (2022) train transformer models on search trajectories to mimic search strategies such as Monte Carlo Tree Search or BFS. While their approach focuses on mimicking a single fixed search procedure, we are interested in autonomous usage of different search procedures and the discovery of new ones. In recent concurrent work, Lehnert et al. (2024) employ A* search traces to train a base transformer model, where each trace includes the state and fixed A* heuristic value and search procedure. Their work aims to train transformers to closely imitate A*, and enhance its efficiency. In contrast, our method emphasizes *discovery* of new search strategies. We let the model learn its strategies, with the choice of what to keep implicit vs explicit in the search process shaping what the model improves upon.

A related line of research, starting with Daumé et al. (2009) (*c.f.* Ross et al. (2011); Ranzato et al. (2015)), suggests supervising the process of finding a solution into a structured prediction problem. This is similar in spirit to SoS where there is supervision over the process of search. While Daumé et al. (2009) provides a theoretical framework for learning a policy over structured 'process' data, SoS demonstrates how the search process itself can be represented as a flat language sequence, defining an explicit vocabulary to represent elements of search algorithms, such as backtracking and heuristics, allowing LMs to directly learn diverse search strategies from data.

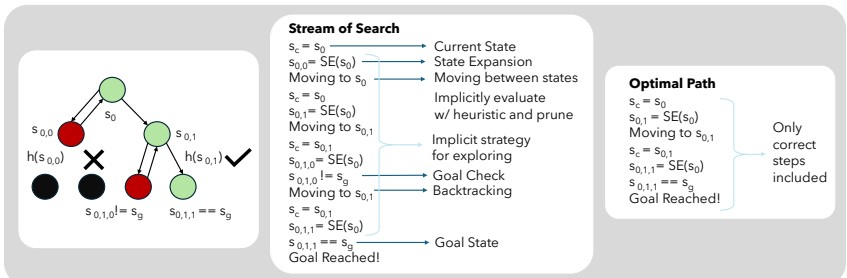

Figure 2: A visualization of how a search process is translated into a stream of search with a language for search. (left) The search process represented as a tree, with different states and operations. The colored states represent the search trajectory $\mathcal{T}$, the green states represent the correct path to the goal $\mathcal{P}$ and the arrows represent transitions between the states. The black circles represent unexplored states. (center) The search process serialized as text to create a stream of search. The labels specify the different components of the process. See Fig. 1b for how this is realized in Countdown. (right) The optimal path, $\mathcal{P}$, to the goal state. Backtracking, exploration and the messy process of search are excluded.

## 3 A Language for Search

The problem space can be modeled as a Markov Decision Process (MDP), with a set of states, $\mathcal{S}$, representing the steps in solving a problem; a set of actions $\mathcal{A}$, representing operations that the model can perform on states or to transition between them; a transition function $T : \mathcal{S} \times \mathcal{A} \to \mathcal{S}$, defining the transition from one state to another based on an action; and a reward function $R : \mathcal{S} \to \mathbb{R}$, assigning a reward for reaching the goal state.

The process of search can be modeled as follows. The query for search has an initial state $s_0 \in \mathcal{S}$ and a goal state $s_g \in \mathcal{S}$. The search tree is defined by the problem itself. It contains all possible explorations of all actions from the initial state $s_0$ (containing the input) to each possible child state of $s_0$, and so on until a leaf state (a state with no children) is reached. A correct path to the solution $\mathcal{P}$ is a sequence of states and actions $(s_0, a_0, s_1, a_1, \ldots, s_{g-1}, a_{g-1}, s_g)$ in the search tree, where each successive state $s_{i+1}$ is obtained by applying a valid action $a_i \in \mathcal{A}$ to the previous state $s_i$, i.e., $s_{i+1} = T(s_i, a_i)$, and the final state in the sequence is the goal state $s_g$ (see Fig. 2).

We are interested in the process of search, which corresponds to moving around the search tree until a solution is found. To represent this process, we propose a vocabulary of primitive operations[2] that can be used to define components of different search algorithms. This is in spirit similar to defining a domain-specific language for planning (Fikes & Nilsson, 1971).

- **Current State:** $s_c$, The state $s_c$ that is being explored.
- **Goal State:** $s_g$, The state $s_g$ that is the target.
- **State Queue:** $S_q$ The states at the 'frontier' of the the the trajectory that haven't been explored yet.
- **State Expansion Function:** $SE : \mathcal{S} \to \mathcal{S}$, Explore a state adjacent to the current state $s_c$ based on a transition function $T$.
- **Exploration Choice:** Choosing the order of states to explore following the state expansion. Eg: Should the breadth be explored first (BFS) or the depth (DFS) or any of the frontier states ($A^*$).
- **Pruning:** An action discarding states or subtrees that are unlikely to lead to a solution.
- **Backtracking:** An action to move between nodes that have been explored. This allows the algorithm to choose which state should be expanded next.
- **Goal Check:** An action to check if the current state is the goal state ($s_c == s_g$).
- **Heuristic:** A function $h \in \mathcal{H} : \mathcal{S} \times \mathcal{S} \to \mathbb{R}$ that approximates the distance of the current state $s_c$ from the goal $s_g$. This can be used to decide which states should be explored or pruned. The heuristic function serves as an approximation of the

---

[2]This is a non-exhaustive list, we choose to represent the most salient operations here.

value function, guiding the search or decision-making process by estimating the desirability of states in terms of their distance to the goal.

Each of these operations can be left implicit, affecting how the trajectory unfolds, or made explicit in language as part of the search trajectory $\mathcal{T}$. When operations are implicit, a model is more likely to internalize abstract representations for them that can be improved with training. Explicit operations will turn into explicit reasoning moves made by the LM. We choose to represent the current state, the goal state, the backtracking operations, the goal checks and the exploration choices explicitly in the trajectory (written using language, Fig. 2). We choose to keep the heuristic functions, values of states and the pruning strategy implicit.

## 4    Problem Setup

**Task Description: Countdown**    To show the utility of using streams of search, we look at a generalization of the 24 Game (Yang et al., 2022) called Countdown (Countdown, 2024). Countdown is a game where a set of input numbers need to be combined with simple arithmetic operations to reach a target number (see Fig. 1a). We choose this task, since it has a high branching factor ($\binom{N}{2} * 4$ for a depth that has N inputs) and thus requires planning, search and backtracking to be solved. We consider problems with 4 input numbers since these problems are challenging enough to have long search traces without the search traces exceeding a standard LM context window (for example, the search trajectories for games with 5 input numbers can be 60,000 tokens long). The range of target numbers in our problems is from 10 to 100. We randomly hold out 10% of the targets for an 'out-of-distribution' evaluation.

**Training data**    To train a model on streams of search for Countdown, we construct a synthetic dataset using set of diverse, and suboptimal symbolic search strategies. To construct our Stream of Search dataset, we define 12 search strategies based on breadth first search (BFS) and depth first search (DFS) (see App. Alg. 4, Alg. 3) that rely on two simple and interpretable heuristic functions. The heuristics we use to guide search are 1) the absolute difference between the sum of the remaining options and the target and 2) distance to the factors of the target (see App. B). We generate a stream of search dataset of 500,000 search trajectories and corresponding trajectories with just the optimal solution. Of these 500,000 trajectories, only about 57% of the trajectories lead to the solution. In our dataset, a search trajectory or a stream of search is serialized as a string that represents a list of tree nodes / states in the order of traversal (either generation or exploration). We hold out two kinds of generalization cases: 1) seen targets with new sets of inputs, and 2) new targets with new sets of inputs.

**Metrics**    To measure accuracy in Countdown, we evaluate the percentage of problems for which the model is able to generate a correct solution trajectory. More formally, we define correctness as a binary function, $\mathbf{1}$, if the correct path to the solution, $\mathcal{P}$, is present in the generated trajectory, $\mathcal{T}$.

To quantitatively understand the search strategies used by the trained models, we define two ways to measure alignment between different search strategies:

1. Alignment of Correctness: This measures whether two search strategies solve the same set of problems correctly and incorrectly. We calculate this as the Pearson correlation between the solved and unsolved problems for the two search strategies.
2. Alignment of States Visited: This measures the overlap in the states visited by two search strategies. To calculate this, we parse the search trajectories, $\mathcal{T}_1$ and $\mathcal{T}_2$, into their constituent states and count the number of states that are common between them. We normalize this count by dividing it by the maximum number of states in the two trajectories. More formally: State Alignment$(\mathcal{T}_1, \mathcal{T}_2) = \frac{|\mathcal{T}_1 \cap \mathcal{T}_2|}{\max(|\mathcal{T}_1|, |\mathcal{T}_2|)}$ where $|\mathcal{T}_1|$ and $|\mathcal{T}_2|$ denote the number of states in trajectories $\mathcal{T}_1$ and $\mathcal{T}_2$, respectively. To measure alignment between two models, we calculate the state alignment score for each problem and then compute the mean of these scores.

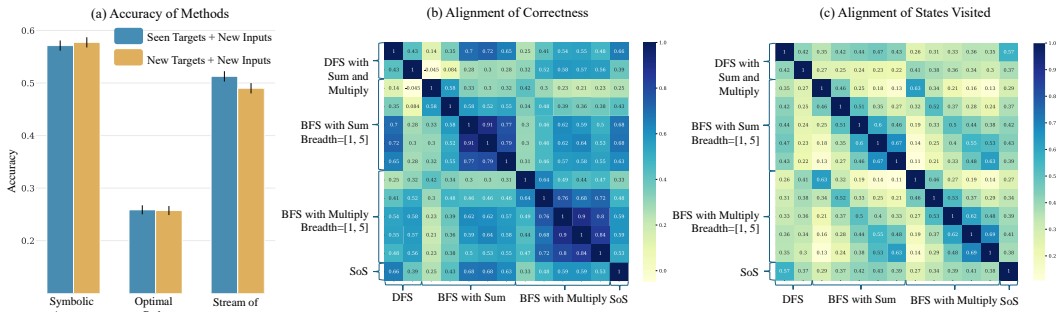

Figure 3: (a) Average accuracy of the symbolic search strategies used to construct the Stream of Search dataset compared to the model trained with optimal paths and the Stream of Search model. Error bars represent 95% binomial confidence intervals. (b) Alignment of Correctness: The correlations of performance on different problems for the Stream of Search model and different symbolic search strategies. (c) Alignment of States Visited: Number of times each search strategy visited the same states, normalized by the maximum number of states in the two trajectories. We find that the alignment scores for states visited for the SoS model are not highly correlated with any single symbolic strategy.

## 5    Learning from Suboptimal Search Strategies

Is it more useful to learn from clean, optimal solutions, or from messy, and sometimes unsuccessful, search trajectories? In this section we train LMs from scratch on one or the other and evaluate there performance at solving held out Countdown problems.

**Experiment Setup.** For all experiments, we train a GPT-Neo model (Gao et al., 2020) with 250M parameters and a context length of 4096 tokens (see appendix for more details). We train models with a causal language modeling objective on the 500,000 Countdown problems in two conditions: 1) Optimal Paths (OP): the model is trained to predict the correct, and optimal path $\mathcal{P}$ for all problems in the dataset. 2) Stream of Search (SoS): The model is trained on search trajectories $\mathcal{T}$ sampled from different search strategies. The SoS dataset has 285,000 correct solutions while the OP dataset has 500,000 correct solutions. Both models are trained up to the same number of gradient steps.

**Results.** We find that the model trained on streams of search outperforms the model trained on the optimal solutions (see Fig. 3a). The SoS model achieves an accuracy of 51.27% on held-out inputs compared to 25.73% of the OP model. A similar pattern is seen for the test set with held-out targets. This shows that despite having fewer 'correct' examples for training, the SoS model outperforms the OP model. The SoS model has a slightly lower accuracy compared to the average accuracy of the symbolic search methods used to construct the SoS training dataset. However, it is worth noting that SoS is solving a harder problem: symbolic search relies on an environment model to expose state transitions, while the SoS LM simulates those transitions itself, effectively learning to search with an internal 'world model'; the language model must also learn how to perform arithmetic operations. Despite these challenges, the SoS model generates valid trajectories with a low error rate in state exploration (0.8%) and only about 2 arithmetic errors per trajectory on average (see App. Tab. 2). x

To understand the strategies that the trained SoS model uses, we measure the alignment of the model generated search trajectories with symbolic strategies. We find that the alignment scores for states visited for the SoS model are not highly correlated with any single symbolic strategy (see Fig. 3c). The highest correlation of SoS is with DFS using the sum heuristic (0.57), and the lowest correlation is with BFS using a breadth size of 5 and the sum heuristic (0.27). A similar pattern is observed when we measure the alignments of correctness (see Fig. 3b) for the SoS model and the symbolic search strategies. Overall, the SoS model has higher scores for alignment with strategies that use the sum heuristic, but does not seem to predominantly use any one strategy from its training data.

## 6 Policy Improvement with Stream of Search

The SoS LM has learned to use search to solve new problems. Can it learn to improve upon the symbolic strategies seen in its training data? In this section, we explore if the model can self-improve with feedback based on correctness and efficiency. We measure ability to solve previously unsolved problems (by the symbolic search strategies) from the training dataset and solve difficult problems from the training set that none of the symbolic search strategies can solve. To improve the model, we use two RL strategies, expert iteration using STaR (Zelikman et al., 2022), and Advantage-Induced Policy Alignment (Zhu et al., 2023).

**Experiment Setup.** To improve the models with STaR (Alg. 1), we use problems in the training dataset to generate 100,000 correct trajectories. We sample with a temperature of 0.8. These trajectories are then used to finetune the model. We repeat this process until we see a convergence in performance on the validation set.

Alternatively, we can use advantage-induced policy alignment (APA; Alg. 2). APA is an Actor-Critic reinforcement learning technique that involves creating a copy of the language model to serve as a value network that then used to enhance the policy, the original language model. We define a straightforward reward function that takes into account the correctness and length of the generated trajectory. We try APA, as we wanted to see if using a separate value network would improve exploration and the path-stitching abilities of the language model. We chose APA over other methods like Proximal Policy Optimization (PPO) due to its stability and robustness to changes in hyperparameters.

APA uses a reference policy, $\pi_{ref}$, to prevent the policy from drifting from its initial state. We observe that updating the reference policy whenever the validation reward converges results in further policy improvement (see Fig. 4b). This shifting of the reference distribution can be interpreted as a means to reduce the weight assigned to staying close to the reference distribution (the $\lambda$ parameter in the APA objective). In practice, we found the strategy of shifting the reference distribution to be more stable for training when compared to designing a schedule for reducing $\lambda$ over training.

---

**Algorithm 1** Expert Iteration with STaR (Zelikman et al., 2022)

---

1: **Input** The SoS model trained on Stream of Search Dataset $D = \{(x_i, \mathcal{T}_i)\}_{i=1}^{m}$
2: $M_0 \leftarrow SoS$            ▷ Initialize the SoS model
3: **for** $n$ in $1 \ldots N$ **do**
4:      $\mathcal{T}_i \leftarrow M_{n-1}(x_i) \; \forall i \in [1, m]$          ▷ Perform trajectory generation
5:      $D_n \leftarrow \{(x_i, \mathcal{T}_i) | i \in [1, m]\} \text{ s.t. } \mathcal{P}_i \in \mathcal{T}_i$      ▷ Filter trajectories based on correctness
6:      $M_n \leftarrow \text{train}(M_0, D_n)$          ▷ Finetune the model on correct solutions
7: **end for**

---

**Algorithm 2** Advantage Indued Policy-Alignment (APA) (Zhu et al., 2023)

---

1: **Input:** An initial policy parameter $\pi_{init}$, a given reward function $R$, Advantage coefficient $\lambda$.
2: $\pi_0 \leftarrow \pi_{init}$
3: $\pi_{ref} \leftarrow \pi_{init}$          ▷ Copy the SoS model to create a reference network.
4: $\pi_{value} \leftarrow \pi_{init}$          ▷ Copy the SoS model to create a value network.
5: **for** $t$ in $1 \ldots T$ **do**
6:      Roll out $\pi_{\theta_{t-1}}$ to produce dataset $D_t = \{(s_1^{(t)}, a_1^{(t)}, r_1^{(t)}), \cdots, (s_n^{(t)}, a_n^{(t)}, r_n^{(t)})\}$
7:      Update policy function according to
8:          $\theta_t = \arg\max_\theta \mathcal{L}_{APA}(\theta; D_t).$      ▷ We omit the critic loss for simplicity
9:      where
10:          $\mathcal{L}_{APA}(\theta; D) = \frac{1}{|D|} \sum_{(s,a) \in D} \left( \log \pi_\theta(a|s) - \frac{Adv^{\pi_{\theta_{t-1}}}(s,a)}{\lambda} - \log \pi_{ref}(a|s) \right)^2.$
11:      If validation reward converges, update $\pi_{ref}$
12:          $\pi_{ref} \leftarrow \pi_{\theta_t}$
13: **end for**

---

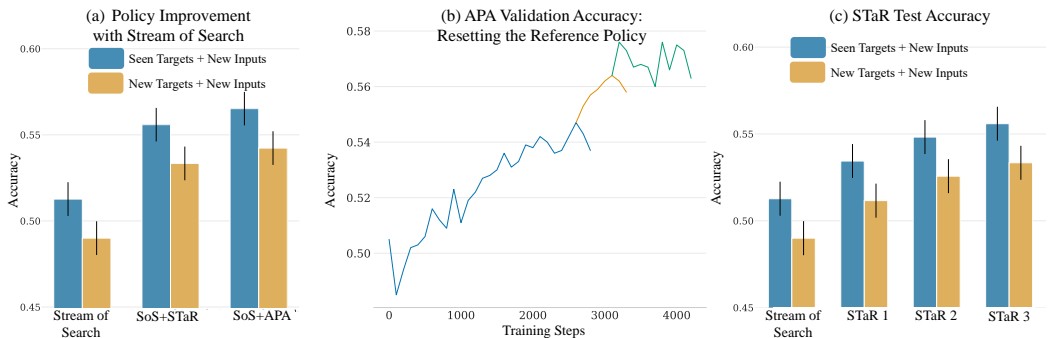

Figure 4: (a) Improvement in accuracy with different policy improvement methods. (b) Accuracy on the validation set of problems during APA training. We reset the reference policy after the validation accuracy converges. Different colors represent training after resetting the reference policy to the current model. We stop seeing improvements after 3 such resets. (c) Improvements in test accuracies with each STaR iteration. Error bars represent 95% Binomial Confidence Intervals.

**Results.** The SoS models converge after 3 iterations of STaR finetuning (Fig. 4). We see that after three iterations, the finetuned SoS+STaR model solves an additional 5% of the held-out inputs test set beyond the base SoS model. A similar pattern is seen for the held-out targets test set. When SoS models are finetuned with APA, we see that the validation accuracy stops improving after about 4000 training steps. We reset the reference policy 3 times, when the validation accuracy stops improving — see Fig. 4b; different colors represent training after resetting the reference policy. Overall, we see an improvement of about 6% over the base SoS model.

When we analyze the difference between the base and finetuned models in terms of the alignment of the states visited, we observe (Fig. 5a) that both the STaR and APA models visit more states associated with the 'multiply' heuristic, which measures distance to the factors of the target. Further, we note that the APA model is less aligned with the symbolic strategies compared to the base SoS model, indicating that it diverges more from the symbolic strategies and employs different strategies for searching. These state visitation metrics provide insights into how the SoS+STaR and SoS+APA models can flexibly utilize various search strategies, potentially discovering novel heuristics and search methods.

To further evaluate the performance of the improved models, we select 10,000 problems from the SoS training set that were unsolved by symbolic strategies when the dataset was generated, and 10,000 difficult problems that none of the symbolic strategies used to train the SoS models can solve. Remarkably, the models are able to solve approximately 36% of the previously unsolved problems (Fig. 5b) and about 4% of the difficult problems (Fig. 5c). Finally, SoS + APA and SoS + STaR models also have better models of the environment, making fewer errors while searching (Fig. 6; App. Tab. 2), and finding the solution more quickly (Fig. 6).

# 7  Discussion

We have introduced the Stream of Search (SoS) framework enabling language models to learn to solve problems by searching in language, without any external structure or components. By systematizing the elements of search into a unified language, we are able to represent various search strategies in a common format to construct a dataset with diverse streams of search. Our experiments demonstrate that training language models to search leads to superior performance compared to models trained solely on optimal trajectories. This highlights the importance of exposing models to the messy process of problem solving, with exploration and backtracking, instead of only the ideal solution steps. SoS models can then self-improve by optimizing for correctness, using STaR and APA.

The SoS framework may address criticisms (LeCun, 2023; Bachmann & Nagarajan, 2024) of language models for planning and problem solving. The problem of snowballing errors is

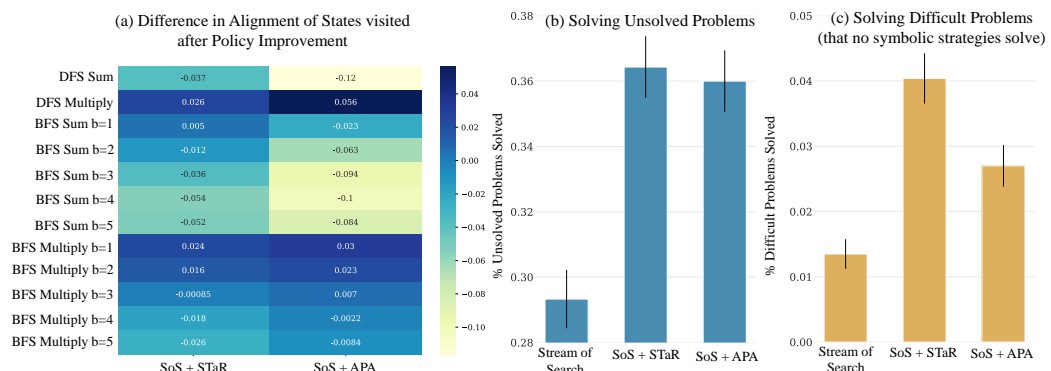

Figure 5: (a) Difference in the states visited compared to the base Stream of Search Model after finetuning with APA and STaR. We see that the model prefers the states visited by certain symbolic strategies after improvement. (b) The percentage of previously unsolved problems from the training set that the trained models were able to solve. (c) The percentage of difficult problems, defined as those not solvable by any symbolic algorithms, that the trained SoS models successfully solved. Error bars are 95% Binomial Confidence Intervals.

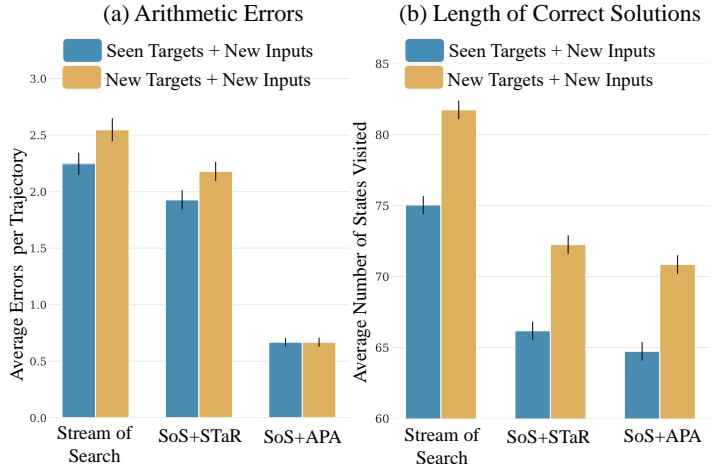

Figure 6: (left) Average number of arithmetic errors made per search trajectory by different models. Policy improvement leads to fewer errors. (right) Average number of states visited per trajectory for correct solutions. Policy improvement leads to more efficient solutions.

addressed by teaching a model to backtrack. Search allows models to explore alternative paths, overcoming failures in lookahead tasks by considering multiple possible outcomes before committing to a course of action. Crucially, SoS leads language models to learn an internal 'world model' for search. Unlike symbolic search that relies on an explicit environment model, SoS models simulate state transitions themselves. Using a learned world models allows more adaptable and generalizable search (Cf. Schrittwieser et al., 2020) and addresses a key criticism of pretrained LMs.

Our empirical results were restricted to the game of Countdown. Yet Countdown, with a high branching factor and variable goal states, captures the characteristics of complex planning problems. We are optimistic that SoS extends to more challenging, real-world tasks. In the short-term, externally-structured search methods such as Tree of Thought (Yao et al., 2024) are likely to be more efficient for these tasks; in the longer run the increased flexibility and learnability of internally-structured search (i.e. SoS) may prevail. It is also plausible these approaches can be combined, for sinstance, with with ToT providing initial training data for SoS.

While we leave the evaluation of states to be done implicitly by the network in our current work, explicitly representing state evaluations (Gandhi et al., 2023) and introducing other formalizable operations such as limits, summarization, cycle checks, and subgoal setting could enhance the SoS framework. Future research could explore integrating subgoals and hierarchical planning, as well as incorporating reflection and self-evaluation to enable models to discover and improve novel search strategies (Huang et al., 2023; Shinn et al.,

2024; Stechly et al., 2024). Generating the initial SoS dataset can be challenging, as it is not always feasible to create symbolic search algorithms to solve problems. An important question is how well search abilities transfer between domains and between formal and informal domains.

In conclusion, we have shown that hallmarks of symbolic reasoning—structured search with backtracking, heuristic state evaluation and world modeling—can be achieved within a sequence modeling paradigm. To do so requires showing language models examples of productive mistakes, not only the optimal final solutions. By embracing the diversity of search strategies and iteratively refining models, we can unlock the potential of language models to tackle complex problems and discover new ways to solve them.

## Acknowledgements

We would like to thank Gabriel Poesia, Jacob Andreas, Joy He-Yueya, Dongwei Jiang, Joseph Feffer, Eric Zelikman, Jan-Philipp Fränken and Ced Zhang for their discussions and support. This worked was supported by the Stanford Human-Centered Artifical Intelligence (HAI) - Google grant, and the NSF Expeditions Grant, Award Number (FAIN) 1918771.

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

## A  Reproducibility Statement

We release the code to replicate our experiments here: https://github.com/kanishkg/stream-of-search. We use the APA implementation available here: https://github.com/microsoft/RLHF-APA/tree/main/examples. We optimize their code for parallelism, memory, mixed precision training and to use flash-attention (Dao et al., 2022). See our training fork here: https://github.com/kanishkg/RLHF-APA.

## B  Creation of the SoS Dataset

**SoS Dataset.**To generate the Stream of Search dataset, we use two symbolic strategies, BFS (Alg. 4) and DFS (Alg. 3). These strategies are guided by two heuristic functinos to generate trajectories to create the SoS training set. Let the input numbers for the Countdown problem be $I = [n_1, n_2, \ldots n_k]$, and the target number be $T$. The sum heuristic measures the distance between the sum of all input numbers to the target number: $h_{sum}(I, T) = |T - \sum_{i=1}^{i=k} n_i|$. The multiply heuristic measures the minimum distance between the sum of inputs and the factors of the target. If $T$ has factors $[f_1, f_2 \ldots f_m]$. So, the multiply heuristic can be written as $h_{multiply} = \min([|f_j - \sum_{i=1}^{i=k} n_i|] \forall j \in [1, m])$. For DFS, the threshold for the heuristic is set to the value of the target $h_{th} = T$. For BFS, a breadth limit $b \in [1, 2, 3, 4, 5]$ is used to keep only the top $b$ branches each state expansion. These hyperparameters provide a simple and intuitive way to design symbolic algorithms to generate the SoS dataset.

The SoS dataset has 285,501 (57.1%) correct trajectories.

**Optimal Paths dataset.** The optimal trajectories only include the correct steps to reach the goal state. There are 500,000 correct solutions shown to the optimal paths model.

---

**Algorithm 3** SoS-DFS

---

**Require:** Problem $x$, Heuristic $h$ State Expansion Function $SE$, threshold $h_{th}$, Goal State $s_g$.
  SoS = ""
  $s_c \leftarrow x$
  **if** $s_c == s_g$ **then**
    **return** SoS + "Goal Reached"
  **end if**
  **if** $G(s_c)$ is Null and $s_c \mathrel{!=} s_g$ **then**
    **return** SoS
  **end if**
  $S_q \leftarrow s \ \forall \ s \in SE(s_c)$ s.t. $h(s) > h_{th}$
  **for** $s \in S$ **do**
    SoS += $s$
    SoS += DFS($s$)
  **end for**
  **return** SoS

---

**Optimal Path Solution**

```
Current State: 18:[74, 24, 36, 44], Operations: []
Exploring Operation: 74+24=98, Resulting Numbers: [36, 44, 98]
Generated Node #2: [36, 44, 98] from Operation: 74+24=98
Current State: 18:[36, 44, 98], Operations: ['74+24=98']
Exploring Operation: 36+44=80, Resulting Numbers: [98, 80]
Generated Node #3: [98, 80] from Operation: 36+44=80
Current State: 18:[98, 80], Operations: ['74+24=98', '36+44=80']
Exploring Operation: 98-80=18, Resulting Numbers: [18]
18,18 equal: Goal Reached
```

Figure 7: An example of an Optimal Path to the solution.

---

**Algorithm 4** SoS-BFS

---

**Require:** Problem $x$, Heuristic $h$, breadth limit $b$, Next State Expander $SE$, Goal State $s_g$
  $S_{q,0} \leftarrow x$
  SoS = ""
  **for** $i = 1, \ldots, T$ **do do**
    $s_c \leftarrow s_{i-1}$ where $s_{i-1} \in S_{q,i-1}$
    SoS += $(s_c)$
    **if** $s_c == s_g$ **then**
        return SoS += "Goal Reached"
    **end if**
    $S_i \leftarrow SE(s_c)$
    $H_i \leftarrow h(S_i)$
    $S_{q,i} \leftarrow \arg\max_{S \subseteq S'_i, |S|=b} H_i(s)$
    SoS += $(S_{q,i})$
  **end for**
  **return** SoS

---

## C  Training Details

Models are trained with an Adam Optimizer on 4 x 80 GB A100 GPUs. For all methods, model selection is done based on accuracy on a validation set of 1,000 problems. All hyperparameters in this section are per GPU unless specified otherwise.

### C.1  Model Architecture

We train a GPT-Neo model (Gao et al., 2020) with the following architecture:

- Layers: 16 (all global attention)
- Heads: 16
- Context length: 4096
- Hidden Size: 1024
- Dropout: 0.1
- Data Type: `bf16`
- Attention: Flash-Attention-2 (Dao et al., 2022)

### C.2  Stream of Search Model

- Number of training examples: $5 \times 10^5$
- Evaluation steps: 500
- Batch size: 24
- Gradient accumulation steps: 1
- Number of training steps: 50,000
- Weight decay: 0.01
- Warmup steps: 1
- Learning rate scheduler type: Cosine
- Learning rate: $1 \times 10^{-5}$

### C.3  Optimal Paths Model

- Number of training examples: $5 \times 10^5$
- Batch size: 24
- Gradient accumulation steps: 1
- Number of training steps: 50,000
- Weight decay: 0.01
- Warmup steps: 1

**Stream of Search Solution**

```
Current State: 18:[74, 24, 36, 44], Operations: []
Exploring Operation: 74-44=30, Resulting Numbers: [24, 36, 30]
Generated Node #0,0: 18:[24, 36, 30] Operation: 74-44=30
Exploring Operation: 44-36=8, Resulting Numbers: [74, 24, 8]
Generated Node #0,1: 18:[74, 24, 8] Operation: 44-36=8
Exploring Operation: 44-24=20, Resulting Numbers: [74, 36, 20]
Generated Node #0,2: 18:[74, 36, 20] Operation: 44-24=20
Exploring Operation: 74-24=50, Resulting Numbers: [36, 44, 50]
Generated Node #0,3: 18:[36, 44, 50] Operation: 74-24=50
Exploring Operation: 74-36=38, Resulting Numbers: [24, 44, 38]
Generated Node #0,4: 18:[24, 44, 38] Operation: 74-36=38
Moving to Node #0,0
Current State: 18:[24, 36, 30], Operations: ['74-44=30']
Exploring Operation: 36-24=12, Resulting Numbers: [30, 12]
Generated Node #0,0,0: 18:[30, 12] Operation: 36-24=12
Exploring Operation: 36-30=6, Resulting Numbers: [24, 6]
Generated Node #0,0,1: 18:[24, 6] Operation: 36-30=6
Exploring Operation: 24+30=54, Resulting Numbers: [36, 54]
Generated Node #0,0,2: 18:[36, 54] Operation: 24+30=54
Exploring Operation: 30-24=6, Resulting Numbers: [36, 6]
Generated Node #0,0,3: 18:[36, 6] Operation: 30-24=6
Exploring Operation: 24+36=60, Resulting Numbers: [30, 60]
Generated Node #0,0,4: 18:[30, 60] Operation: 24+36=60
Moving to Node #0,4
Current State: 18:[24, 44, 38], Operations: ['74-36=38']
Exploring Operation: 44-38=6, Resulting Numbers: [24, 6]
Generated Node #0,4,0: 18:[24, 6] Operation: 44-38=6
Exploring Operation: 44-24=20, Resulting Numbers: [38, 20]
Generated Node #0,4,1: 18:[38, 20] Operation: 44-24=20
Exploring Operation: 24+38=62, Resulting Numbers: [44, 62]
Generated Node #0,4,2: 18:[44, 62] Operation: 24+38=62
Exploring Operation: 24+44=68, Resulting Numbers: [38, 68]
Generated Node #0,4,3: 18:[38, 68] Operation: 24+44=68
Exploring Operation: 38-24=14, Resulting Numbers: [44, 14]
Generated Node #0,4,4: 18:[44, 14] Operation: 38-24=14
Moving to Node #0,1
Current State: 18:[74, 24, 8], Operations: ['44-36=8']
Exploring Operation: 74-8=66, Resulting Numbers: [24, 66]
Generated Node #0,1,0: 18:[24, 66] Operation: 74-8=66
Exploring Operation: 24-8=16, Resulting Numbers: [74, 16]
Generated Node #0,1,1: 18:[74, 16] Operation: 24-8=16
Exploring Operation: 24+8=32, Resulting Numbers: [74, 32]
Generated Node #0,1,2: 18:[74, 32] Operation: 24+8=32
Exploring Operation: 74+8=82, Resulting Numbers: [24, 82]
Generated Node #0,1,3: 18:[24, 82] Operation: 74+8=82
Exploring Operation: 74-24=50, Resulting Numbers: [8, 50]
Generated Node #0,1,4: 18:[8, 50] Operation: 74-24=50
Moving to Node #0,2
Current State: 18:[74, 36, 20], Operations: ['44-24=20']
Exploring Operation: 74+20=94, Resulting Numbers: [36, 94]
Generated Node #0,2,0: 18:[36, 94] Operation: 74+20=94
Exploring Operation: 74-20=54, Resulting Numbers: [36, 54]
Generated Node #0,2,1: 18:[36, 54] Operation: 74-20=54
Exploring Operation: 36-20=16, Resulting Numbers: [74, 16]
Generated Node #0,2,2: 18:[74, 16] Operation: 36-20=16
Exploring Operation: 74+36=110, Resulting Numbers: [20, 110]
Generated Node #0,2,3: 18:[20, 110] Operation: 74+36=110
Exploring Operation: 74-36=38, Resulting Numbers: [20, 38]
Generated Node #0,2,4: 18:[20, 38] Operation: 74-36=38
Moving to Node #0,3
Current State: 18:[36, 44, 50], Operations: ['74-24=50']
Exploring Operation: 44-36=8, Resulting Numbers: [50, 8]
Generated Node #0,3,0: 18:[50, 8] Operation: 44-36=8
Exploring Operation: 36+44=80, Resulting Numbers: [50, 80]
Generated Node #0,3,1: 18:[50, 80] Operation: 36+44=80
Exploring Operation: 50-36=14, Resulting Numbers: [44, 14]
Generated Node #0,3,2: 18:[44, 14] Operation: 50-36=14
Exploring Operation: 50-44=6, Resulting Numbers: [36, 6]
Generated Node #0,3,3: 18:[36, 6] Operation: 50-44=6
Exploring Operation: 36+50=86, Resulting Numbers: [44, 86]
Generated Node #0,3,4: 18:[44, 86] Operation: 36+50=86
Moving to Node #0,0,0
Current State: 18:[30, 12], Operations: ['74-44=30', '36-24=12']
Exploring Operation: 30-12=18, Resulting Numbers: [18]
18,18 equal: Goal Reached
```

Figure 8: An example of a Stream of Search trajectory.

- Learning rate scheduler type: cosine

- Learning rate: $1 \times 10^{-5}$

### C.4 Stream of Search + STaR

For STaR, we use a temperature of 0.8 to sample trajectories for problems in the training set. We filter search trajectories so that only correct trajectories are used to finetune the next iteration. Similar to Zelikman et al. (2022) we reset the model to the base SoS model at each iteration of improvement.

- Number of training examples: $1 \times 10^5$
- Batch size: 24
- Gradient accumulation steps: 1
- Number of training steps: 20,000
- Weight decay: 0.01
- Warmup steps: 100
- Learning rate scheduler type: cosine
- Learning rate: $1 \times 10^{-5}$

### C.5 Stream of Search + APA

- Number of rollouts: 32
- Temperature for sampling: 1.0
- Batch Size: 8
- Online Epochs: 2
- Gradient accumulation steps: 1
- Scaling of reward: None
- Value Loss Coefficient: 10
- $\gamma$: 1.0
- $\lambda$: 2.0
- Learning rate scheduler type: None
- Learning rate: $1 \times 10^{-6}$

## D Analysis of generated trajectories

In each evaluation of the trained model, we prompt the model to generate trajectories on 1) 10,000 tests with seen targets and unseen sets of input numbers, and 2) 10,000 tests with unseen or held-out targets. On top of improving accuracy, both policy improvement methods also reduced the average number of states or nodes explored before reaching the goal in correct trajectories, compared to the correct trajectories generated by the SoS LM. Under our framework, the number of states explored is correlated with the length of generation. This demonstrates that the policy improvement methods increase the efficiency of search.

To further study the factors contributing to the correctness of trajectories, an error analysis was conducted for each one. Errors are grouped into four main types: 1) arithmetic (hallucinating invalid operations, dividing by zero, etc.), 2) formatting, 3) exploration (jumping to a node that has no parents), and 4) other (a child node has the wrong set of numbers associated with it). We can see that the APA training method suppressed arithmetic errors significantly and preserved a low rate of exploration errors. Meanwhile, the STaR method, while boosting accuracy, had an increased rate of exploration anomalies and other memory-based discrepancies.

We also analyze the correctness of solutions as a function of solution length in Tab. 3. We find that accuracy scores remain similar across most lengths, suggesting no significant degradation. All 4096-token trajectories are incorrect.

| Model | Target type | Accuracy | Num. states explored in correct trajectories |
|-------|-------------|----------|----------------------------------------------|
| SoS | Seen | 0.5127 | $75.043 \pm 0.640$ |
| SoS | Unseen | 0.4900 | $81.747 \pm 0.643$ |
| STaR | Seen | 0.5559 | $66.176 \pm 0.641$ |
| STaR | Unseen | 0.5334 | $72.252 \pm 0.647$ |
| APA | Seen | 0.5652 | $64.733 \pm 0.652$ |
| APA | Unseen | 0.5423 | $70.849 \pm 0.655$ |

Table 1: Details on statistics for each trained model (average over 10000 tests). Error margins represent 95% confidence intervals.

| Model | Target type | Arithmetic | Formatting | Exploration | Other |
|-------|-------------|------------|------------|-------------|-------|
| SoS | Seen | $2.247 \pm 0.097$ | $0.014 \pm 0.003$ | $0.008 \pm 0.003$ | $0.018 \pm 0.005$ |
| SoS | Unseen | $2.547 \pm 0.102$ | $0.014 \pm 0.002$ | $0.005 \pm 0.002$ | $0.011 \pm 0.003$ |
| STaR | Seen | $1.927 \pm 0.084$ | $0.012 \pm 0.002$ | $0.024 \pm 0.004$ | $0.069 \pm 0.009$ |
| STaR | Unseen | $2.179 \pm 0.084$ | $0.013 \pm 0.002$ | $0.022 \pm 0.004$ | $0.074 \pm 0.009$ |
| APA | Seen | $0.668 \pm 0.036$ | $0.014 \pm 0.003$ | $0.008 \pm 0.003$ | $0.022 \pm 0.004$ |
| APA | Unseen | $0.785 \pm 0.039$ | $0.013 \pm 0.002$ | $0.006 \pm 0.002$ | $0.017 \pm 0.003$ |

Table 2: Error analysis for trained models (average number of each error per trajectory, 10000 tests). Error margins are 95% Confidence Intervals.

| Number of Tokens | SoS | STaRx3 | APA |
|------------------|-----|--------|-----|
| 0-1000 | 2884/2994 | 3304/3373 | 3393/3445 |
| 1000-2000 | 1291/1318 | 1255/1304 | 1219/1273 |
| 2000-3000 | 534/545 | 551/588 | 523/661 |
| 3000-4000 | 389/429 | 419/467 | 388/420 |
| 4000-5000 | 29/4764 | 30/4268 | 29/4201 |

Table 3: Correctness of SoS models as a function of solution length. Numbers represent correct solutions / total number of solutions. Accuracy scores remain similar across most lengths, suggesting no significant degradation.

