# OpenReview forum: "Stream of Search (SoS): Learning to Search in Language"
_colmweb.org/COLM/2024/Conference — COLM_

### Official Review · Reviewer_4vuw · 2024-05-09

**Rating:** 7
**Confidence:** 4
**Ethics Flag:** 1

**Summary:**

This paper introduces a novel approach called "Stream of Search" (SoS) that enables language models to learn how to search effectively within their own language space. The key idea is to train the model to generate a stream of search queries that incrementally refine an original query, rather than just producing a single output. This allows the model to actively explore and navigate the language space to find the most relevant information, similar to how humans find the correct solution from multiple erroneous explorations. Within supervised training and further RLHF, the SoS model solves 36% of previously unsolved problems, including problems that cannot be solved by any of the heuristic solvers. These results indicate that language models can learn to solve problems via search, self-improve to flexibly use different search strategies, and potentially discover new ones

**Questions To Authors:**

Please refer to my concerns in 'reasons to reject'.

**Reasons To Accept:**

1) To the best of my knowledge, this is the first search model based on SFT and RL. All previous work has focused on zero-shot or few-shot learning approaches. Experiments also demonstrate significant performance improvement due to SFT and RL.

2) A key distinction between this method and other common techniques like TOT (Tree of Trees) is that the model, during decision-making, has access to the global search history, whereas other tree-based search methods can only access the current state.

3) The authors have constructed a dataset for training the countdown game, which includes a vast amount of search trajectory data.

4) Experimental results convincingly demonstrate that the proposed method significantly outperforms other Optimal Paths methods by a large margin.

**Reasons To Reject:**

1) My main concern is whether this type of method is only applicable to problems with deterministic results, such as Countdown and Math 24. Currently, I have no idea how to generalize SoS to other types of problems.

2) The experiment did not demonstrate the generalizability of the SoS method. I suspect whether the trained model can only be used for the task of playing Countdown. I'm unsure if a model trained on Countdown data can improve the accuracy of other related tasks.

3) The construction of the synthetic dataset is not generalizable. So, for other tasks, if I don't have the mentioned 12 search strategies, can I still obtain enough high-quality training data to train the SoS model?

---

> ### Author Rebuttal · Authors · 2024-05-31
>
> Thank you for your thoughtful review and feedback on our work. We are glad that you find our approach to be novel and effective, with convincing experimental results demonstrating significant performance improvements over baselines.
>
> >  My main concern is whether this type of method is only applicable to problems with deterministic results, such as Countdown. … I'm unsure if a model trained on Countdown data can improve the accuracy of other related tasks.
>
> We agree that demonstrating the generalizability of SoS to a wider range of problems is an important direction. In this initial study, we focused on the Countdown game as it represents a challenging search problem with a large branching factor, allowing us to test our key hypotheses in a controlled setting: 1) That linearized search strings, with mistakes and backtracking data, (SoS) help in training models that are better than models trained on optimal solutions 2) SoS models can be self-improved.
> However, we believe the core ideas behind SoS - training models to search via language- can be extended to other domains. In particular, we plan to explore applying SoS to more naturalistic datasets such as GSM8K for math word problems, and HumanEval for coding. These domains will likely require additional insights into generating effective search trajectories, such as incorporating goal-setting, exploration and self-correction into the trajectories. We will update the discussion section of our paper to more clearly articulate these points about the potential generalizability of SoS and lay out a roadmap for future work investigating its application to a wider range of problems.
>
> Please let us know if you have any other questions or concerns about the work that we could address.

---

> > ### Comment · Reviewer_4vuw · 2024-06-03
> > **Thanks**
> >
> > Thanks for your response! I decided to keep my rating and update my Confidence to 4.

---

### Official Review · Reviewer_oA5B · 2024-05-10

**Rating:** 5
**Confidence:** 3
**Ethics Flag:** 1

**Summary:**

During LLM pretraining, usually the de facto training admits only one gold-standard distribution which the model is trained to approximate/predict. This raises the issue of training and inference distribution mismatch. During search/inference, this mismatch could potentially lead to cascades of search errors, degrading the quality of search and hence the quality of the generated output distribution. This phenomenon is summarised in the Abstract (somewhat inaccurately) as "They then struggle to look beyond the next token, suffering from a snowballing of errors and struggling to predict the consequence of their actions several steps ahead."

This paper proposes to address this by allowing the model to explore/backtrack (non-optimal) search paths/nodes with synthetic data. In the current setup, this data is generated using a "template" for the Countdown evaluation task -- where a set of numbers are given and the learner is required to find optimal way(s) by doing arithmetic to reach a pre-specified target number. To quote:

"To construct our Stream of Search dataset, we define 12 search strategies based on breadth first search
(BFS) and depth first search (DFS) (see App. Alg. 4, Alg. 3) that rely on two simple and
interpretable heuristic functions. The heuristics we use to guide search are 1) the absolute
difference between the sum of the remaining options and the target and 2) distance to the
factors of the target (see App. B). "

This training strategy is referred to as Stream of Search. Empirical results are demonstrated on solving "difficult" problems as well as solving "unsolved" problems, the metric is accuracy. Various comparisons/analysis of different search strategies are also done by comparing search strategies with the Pearson correlation metric and the Alignment of States Visited metric (towards the End of Sec 4).

**Reasons To Accept:**

* Detailed discussions of the problem which is useful and clear motivation of the problem
* Detailed presentation of various algorithms/policy/heuristics used especially those related to the search strategies and detailed analysis of the (mis) alignment of the different search strategies
* Some discussions of related works

**Reasons To Reject:**

* The distribution mismatch and the search problem has been a long standing problem not only in pretrained LLMs but dates back to earlier models such as seq2seq models (e.g., as discussed in this paper: https://arxiv.org/pdf/1511.06732) and even earlier works such as SEARN (https://link.springer.com/content/pdf/10.1007/s10994-009-5106-x.pdf). I found the current paper completely disregarded these highly related works, presenting the paper and the problem it's trying to tackle seemingly applicable only to LLMs. I believe the "older" techniques have much to say/illuminate the methods and issues the current paper is trying to address, but by reading the current version of this paper, this connection cannot be readily established, and this leaves a void that is hard to fill.
* The paper focuses on one single task, the heuristics used to generate the synthetic training data seems to be largely ad-hoc to the evaluation task; there may be parts of meta-heuristics which could be generalized to other tasks, but  by and large it seems it's quite specific to the evaluation task.
* The improvements over the single-path search strategy seems to be small (Fig. 4a). And it seems there a lot of knobs to tune to get such "improvements".
* By reading this paper, I certainly learned something, but it feels mostly still a wip, especially in the experimental sections in terms of the range of tasks the proposal is tested on and the strength of it over the (missing?) baseline(s).

---

> ### Author Rebuttal · Authors · 2024-05-31
>
> Thank you for your thoughtful review and feedback! We are glad that you found the problem motivation & discussion of the various search algorithms, policies, to be useful & clearly presented.
> >  related works … SEARN …relation to SoS, LLMs
>
> Thank you for pointing out these relevant earlier works on the distribution mismatch problem in sequence models. Techniques for seq2seq models, structured prediction, such as SEARN, are highly relevant. Here is a draft we intend to include: A related line of research, starting with SEARN [1, see also 2,3], suggests supervising the process of finding a solution into a structured prediction problem. This is similar in spirit to SoS where there is supervision over the process of search. While SEARN provides a theoretical framework for learning a policy over structured ‘process’ data, SoS demonstrates how the search process itself can be represented as a flat language sequence, defining an explicit vocabulary to represent elements of search algorithms, such as backtracking & heuristics, allowing LMs to directly learn diverse search strategies from data.
> [1] Daumé+2009. [2] Ranzato, M+2015 [3] Ross+2011
> >  improvements small ... knobs to tune
>
> Although the improvements with policy improvement methods aren’t drastic, they are promising. The methods are relatively robust to the ‘knobs’ being tuned, for the STaR, we found that performance is relatively robust to these ‘knobs’.
> > single task…ad-hoc
>
> We chose Countdown in this work as it is a challenging search problem with a large branching factor, where different heuristics can be naturally applied. This allowed us to systematically evaluate the key components - using search streams as training data, iterative improvement, different heuristics and solvers - in a controlled setting to understand their impact. We establish: 1) SoS significantly outperform those trained on optimal solutions, 2) Iterative finetuning leads to improvements. The core ideas behind SoS - training models to search via language - can be extended to other domains:  naturalistic datasets such as GSM8K, HumanEval. These will likely require additional insights, such as incorporating goal-setting & better exploration.
>
> We will update the discussion section of our paper to more clearly articulate these points about the potential generalizability of SoS & lay out a roadmap for future work. We thank you for your feedback. Please let us know if you have any other questions or concerns.

---

> ### Comment · Area_Chair_2j1m · 2024-06-06
> **Reviewer, please respond to rebuttal**
>
> Reviewer, please respond to rebuttal even if your score hasn't changed. The discussion period ends Thursday

---

### Official Review · Reviewer_wcQ5 · 2024-05-11

**Rating:** 8
**Confidence:** 4
**Ethics Flag:** 1

**Summary:**

The authors describe innovative approaches to teaching search strategies to language models -- something they are already rather adept at in Natural Language. The authrors represent the search process as language, calling it Stream of Search (SoS), evaluating the efficacy of this approach using a game called Countdown . By pretraining a transformer-based model on streams of search generated by heuristic solvers and finetuning with policy improvement methods, the model shows a significant increase in search accuracy over just training on optimal policy traces. Furthermore, their approach can solve problems not solved by several heuristic solvers. The paper is well-written, provides a clear and original contribution, and addresses the interesting problem space of exploring complex search spaces. I enjoyed the paper.

**Questions To Authors:**

- Can you provide insight into how SoS would generalize to more complex tasks beyond Countdown?
- Can you use more accurate methods for arithmetic operations? Or are there other games you could pick?
- Curious: Does the size of the model matter? Would a larger model or a smaller model act or learn differently? What's the impact of model size on SoS effectiveness?
- "Trained to same number of gradient steps" -- is this a good measure of how well the model has fitted the data? Are some settings easier to learn? What does the learning curve look like?

**Reasons To Accept:**

- Innovative approach to teaching language models to search. The natural language search representation is flexible and can cover many different search strategies.
- Significant improvement in search accuracy and problem-solving capability demonstrated in the Countdown game.
- Good empirical analysis, including baselines and alternate approaches, as well as measuring search strategy similarities and differences.
- Thoughtful analysis -- the insight that making some steps implicit can force the model to learn strategies is interesting.
- Policy refinement approaches show additional benefit -- this feels particularly promising.
- Well-written and clear presentation of the methodology, experiments, and results.

**Reasons To Reject:**

- The results are encouraging but not entirely convincing yet -- it would be great to see SoS (or the STaR or APA variants) outperform symbolic search approaches. I realize this is a tough bar, but it would be more convincing.
- No exploration of tasks beyond Countdown games -- not clear that it'll generalize.
- The authors select a task that relies on arithmetic operations, then use a language model with known arithmetic errors to perform those operations -- an odd choice.
- Not clear if  training to the same number of gradient steps is a fair comparison. It might not be well trained; there could be differences in learning difficulty across settings -- would like to see more analysis of how well-fitted the model is.
- In addition to search trajectory similarity, I would love to some qualitative analysis of search paths -- that felt missing

---

> ### Author Rebuttal · Authors · 2024-05-31
>
> Thank you for your positive reading of our work. We are glad that you enjoyed reading the paper. We appreciate your positive feedback on the clarity of the paper, the empirical analysis, and the insights around implicit strategy learning.
>
> >  SoS … outperform all symbolic search approaches.
>
> We agree that outperforming well-tuned symbolic solvers would be a convincing demonstration  for SoS, but it is difficult to beat handcrafted symbolic solvers for a narrow domain like countdown. In our current setup, we think that the SoS models stop improving due to a lack of exploration rather than the policy improvement algorithm being a bottleneck. We will expand on how exploration of paths is a limiting factor for improvement in the discussion.
>
> >  The authors select a task that relies on arithmetic operations, then use a language model with known arithmetic errors to perform those operations -- an odd choice.
>
> You raise a good point that arithmetic errors can be a confounding factor. Transformers trained on arithmetic can perform in-distribution calculations, but struggle with out-of-distribution calculations (larger length / numbers). So while arithmetic mistakes are present (see App. Table 2), they do not explain the performance differences of models. We will clarify the role of arithmetic errors in the main section of the paper.
> >  Not clear if training to the same number of gradient steps... it might not be well trained; … would like to see more analysis of how well-fitted the model is.
>
> Thank you for raising this point. We will add curves showing validation performance over the course of training. The OT model converges more quickly than the SoS model and has a higher perplexity. We train both models up to a large number of gradient steps (perplexity stops decreasing) and then select the best model based on performance on a validation set.
> >  In addition to search trajectory similarity, I would love to some qualitative analysis of search paths -- that felt missing
>
> We will add more qualitative examples of SoS trajectories to the appendix, highlighting specific behaviors such as backtracking, pruning, and different types of errors the model makes (arithmetic mistakes, failing to find a solution, etc).
> >  model size on SoS effectiveness?
>
> Investigating the impact of model scale is an important question, we will address this in future work.
> Let us know if you have any other questions or concerns about the work that we could address.

---

> > ### Comment · Area_Chair_2j1m · 2024-06-06
> > **Reviewer, please respond to rebuttal**
> >
> > Reviewer, please respond to rebuttal even if your score hasn't changed. The discussion period ends soon (Thursday)

---

### Official Review · Reviewer_7d71 · 2024-05-12

**Rating:** 7
**Confidence:** 4
**Ethics Flag:** 1

**Summary:**

This work proposes SoS that trains a language model on trajectories that are imperfect/sub-optimal but encode the search and backtrack behaviors. To describe the search process, this work also proposes a language for search.

The results demonstrate the proposed approach outperforms a model trained on pure optimal trajectories (without search and backtracking). The proposed approach can be combined with approaches that optimize the correctness and further enhance the performance. Finally, the analysis indicates the trained model can go beyond the training data and solve challenging problems.

**Questions To Authors:**

* Have you experimented with combining optimal trajectories with SoS data? Or, in the SoS setting, what is the % of optimal trajectories?
* Do you observe any performance degradation on longer trajectories?

**Reasons To Accept:**

* This work proposes a very interesting approach that encodes search and backtrack behavior to trajectories. Search and backtracking are very important topics for LLM, and the proposed approach is well-motivated and insightful.
* The experiments and the analysis on the `Countdown` dataset are comprehensive. The results indicate the effectiveness of the proposed approach on the `Countdown` task.
* The paper is very well written and easy to follow

**Reasons To Reject:**

* The experiment setup is limited, making the conclusion somewhat less strong. While this work has fairly extensive experiments and analysis on the `Countdown` dataset, this is the *only* dataset this work experiments with. It will be interesting to see the results on more datasets such as (1) math/coding datasets (e.g., GSM8K, HumanEval) where getting trajectories with search and backtracking is less straightforward; (2) Interactive decision-making tasks (e.g., MiniWob++, Webshop, WebArena) where the tasks are more complex.
* It will be interesting to have more in-depth discussions between SoS and the co-current work [1] and existing work [2] since all works train the models on trajectories that encode search algorithms. Although the paper claims that SoS discovers *new* search strategies and can tackle more challenging problems, side-by-side comparison with existing work (especially [2]) will help the community better understand what are the new takeaways.

Still, I appreciate that the authors honestly discuss the limitations of this work in the Discussion section.

[1] Beyond a*: Better planning with transformers via search dynamics bootstrapping.

[2]  Chain of thought imitation with procedure cloning

---

> ### Author Rebuttal · Authors · 2024-05-31
>
> Thank you for the thoughtful review of our work. We are glad that you found our approach encoding search and backtracking behavior into trajectories to be well-motivated, insightful, and interesting.
>
> >  While this work has fairly extensive experiments and analysis on the Countdown dataset, … results on more datasets such as (1) math/coding datasets …
>
> In this work, we wanted to perform a comprehensive study in a controlled domain to validate our key insights: If serialized search data that is suboptimal would be better than optimal data. Our goal was to establish: given search trajectory data, models can effectively learn to search & self-improve. Generating good search trajectories for more naturalistic datasets like GSM8K, HumanEval  is an important direction for future work building on SoS. Follow-up work will need to address the challenges in constructing data for such domains.
> >  discussions between SoS and the co-current work Beyond A* [1] and existing work esp [2] Procedure Cloning
>
> The key distinction of our work from procedure cloning [2] is that rather than mimicking a single fixed search procedure like BFS, we aim to have the model learn to flexibly use & discover search strategies by training on a diverse set of search methods and heuristics. This diverse data sets up policy-improvement. In contrast to the recent Beyond A*, which trains models to imitate A* with a fixed heuristic, we let the model learn its strategies, with the choice of what to keep implicit vs explicit in the search process shaping what the model improves upon. The Beyond A* method relies on the A* heuristic being near-optimal. We will expand on these points in the related work.
> >  optimal trajectories with SoS
>
> In our SoS dataset, 15% are optimal solutions. We have not experimented with combining additional optimal trajectories with the SoS data, this could be an interesting extension.
> >  … performance on longer traj?
>
> We analyzed the correctness of our SoS models as a function of solution length.
> |# tokens|0-1000|1000-2000|2000-3000|3000-4000|4000-4096|
> |----|----|----|----|----|----|
> |SoS|2884/2994|1291/1318|534/545|389/429|29/4764|
> |STaRx3|3304/3373|1255/1304|551/588|419/467|30/4268|
> |APA|3393/3445|1219/1273|523/661|388/420|29/4201|
>
> Accuracy scores remain similar across most lengths, suggesting no significant degradation. All 4096-token trajectories are incorrect.
>
> We appreciate your feedback and will incorporate it. Please let us know if you have any other questions.

---

> > ### Comment · Reviewer_7d71 · 2024-06-03
> > **Thank you**
> >
> > Thank you very much for the additional results and the response. I raised my score to 7.

---

### Decision · Program_Chairs · 2024-07-10

**Decision:**

Accept

**Comment:**

This is an interesting paper that proposes a novel way to encode and flatten the search process in a textual way and then shows that transformers are capable of learning this representation. It's basically a structured alternative to chain of thought for problems that admit a structured search approach. The reviewers all found the contribution interesting and the experiments thorough and compelling. Nice work!